# Vaccination against HBV and HAV as Mode of Hepatitis Prevention among People Living with HIV—Data from ECEE Network Group

**DOI:** 10.3390/vaccines11050980

**Published:** 2023-05-14

**Authors:** Kerstin Aimla, Justyna Dominika Kowalska, Raimonda Matulionyte, Velida Mulabdic, Anna Vassilenko, Natalie Bolokadze, David Jilich, Sergii Antoniak, Cristiana Oprea, Tatevik Balayan, Arjan Harxhi, Antonios Papadopoulos, Botond Lakatos, Marta Vasylyev, Josip Begovac, Nina Yancheva, Anca Streinu-Cercel, Antonija Verhaz, Deniz Gokengin, Gordana Dragovic, Lubomir Sojak, Agata Skrzat-Klapaczyńska

**Affiliations:** 1Department of Infectious Diseases, Tartu University Hospital, 50406 Tartu, Estonia; 2Department of Adults’ Infectious Diseases, Hospital for Infectious Diseases, Medical University of Warsaw, 02-091 Warsaw, Poland; 3Department of Infectious Diseases and Dermatovenerology, Faculty of Medicine, Vilnius University, Vilnius University Hospital Santaros Klinikos, LT-08410 Vilnius, Lithuania; 4Clinic for Infectious Diseases, University Clinical Center Sarajevo, 71000 Sarajevo, Bosnia and Herzegovina; 5City Clinical Hospital of Infectious Diseases, 220002 Minsk, Belarus; 6Infectious Diseases, AIDS and Clinical Immunology Research Center, 0160 Tbilisi, Georgia; 7Department of Infectious Diseases, 1st Faculty of Medicine, Charles University in Prague and Faculty Hospital Bulovka, 18000 Prague, Czech Republic; 8Clinic of the Gromashevsky Institute of Epidemiology and Infectious Diseases, 01001 Kyiv, Ukraine; 9Victor Babes Clinical Hospital for Infectious and Tropical Diseases, 030303 Bucharest, Romania; 10National AIDS Center, 0041 Yerevan, Armenia; 11Department of Infectious Disease, Faculty of Medicine, Medical University of Tirana, 1000 Tirana, Albania; 12HIV Unit, 4th Department of Internal Medicine, School of Medicine, National and Kapodistrian University of Athens, Attikon University Hospital, 12462 Athens, Greece; 13National Instititue of Hematology and Infectious Diseases, National Center of HIV, 1097 Budapest, Hungary; 14Lviv Regional Public Health Center, HIV Unit, 79000 Lviv, Ukraine; 15University Hospital of Infectious Diseases, 10000 Zagreb, Croatia; 16Department for AIDS, Specialized Hospital for Active Treatment of Infectious and Parasitic Diseases, Medical University of Sofia, 1606 Sofia, Bulgaria; 17National Institute of Infectious Diseases “Prof. Dr. Matei Balș”, 021105 Bucharest, Romania; 18Clinic for Infectious Diseases Republic of Srpska Banja Luka, 78000 Banja Luka, Bosnia and Herzegovina; 19Department of Infectious Diseases and Clinical Microbiology, Medical Faculty, Ege University, 35100 Izmir, Turkey; 20Department of Pharmacology, Clinical Pharmacology and Toxicology, Faculty of Medicine, University of Belgrade, 11000 Belgrade, Serbia; 21Department of Infectology and Geographical Medicine, 833 05 Bratislava, Slovakia

**Keywords:** HIV, Hepatitis A, Hepatitis B, Hepatitis C, co-infection, vaccination, Central and Eastern Europe

## Abstract

(1) Background: Viral hepatitis C (HCV) and viral hepatitis B (HBV) are common co-infections in people living with HIV (PLWH). All PLWH should be vaccinated against HBV and hepatitis A (HAV) and treated for HBV and HCV. We aimed to compare testing, prophylaxis and treatment of viral hepatitis in PLWH in Central and Eastern Europe (CEE) in 2019 and 2022. (2) Methods: Data was collected through two on-line surveys conducted in 2019 and 2022 among 18 countries of the Euroguidelines in CEE (ECEE) Network Group. (3) Results: In all 18 countries the standard of care was to screen all PLWH for HBV and HCV both years; screening of HAV was routine in 2019 in 54.5% and in 2022 47.4% of clinics. Vaccination of PLWH against HAV was available in 2019 in 16.7%, in 2022 in 22.2% countries. Vaccination against HBV was available routinely and free of charge in 50% of clinics both in 2019 and 2022. In HIV/HBV co-infected the choice of NRTI was tenofovir-based in 94.4% of countries in both years. All clinics that responded had access to direct-acting antivirals (DAAs) but 50% still had limitations for treatment. (4) Conclusions: Although testing for HBV and HCV was good, testing for HAV is insufficient. Vaccination against HBV and especially against HAV has room for improvement; furthermore, HCV treatment access needs to overcome restrictions.

## 1. Introduction

Viral hepatitis C (HCV) and viral hepatitis B (HBV) are common co-infections in people living with HIV (PLWH) due to the shared routes of transmission [1,2]. In Western Europe and North America where the majority of PLWH are men having sex with men (MSM), viral hepatitis caused by HCV rates are estimated between 5 and 20% [3]. In many Central and Eastern European (CEE) countries where the population of PLWH has a high prevalence of former intravenous drug users (IVDU), HCV co-infection rates are ranging from 40% up to 90% [4,5]. The prevailing route of HBV transmission varies between different geographical areas and national vaccination programs. In Europe heterosexual transmission was the most common mode of transmission in 2021, followed by nosocomial transmission and transmission among MSM. Among chronic cases, mother-to-child transmission and nosocomial transmission dominate [6]. In the CEE region, the rates of reported HIV/HBV co-infection are between 2.3% and 40% [5].

As hepatitis A virus (HAV) is mainly transmitted through the faecal-oral route, it has become a major concern among MSM. In addition, PLWH may experience prolonged HAV viremia, which increases transmission within the community [7,8]. Routine vaccination against HAV is optional and usually available as a paid service. However, vaccination against HBV is recommended and included in the national immunisation schedule in almost all countries in the CEE region [9,10]. It is essential to screen all PLWH for HAV and HBV and vaccinate as early as possible, since vaccine response rate is better in HIV-infected patients presenting with a high CD4+ T cell count and low HIV RNA levels. Several studies have shown that vaccination against HAV and HBV is less immunogenic in PLWH than in the general population [11,12,13]. 

Even though there is no vaccine against HCV, since 2011 as direct-acting antivirals (DAA) were developed, there has been very-well-tolerated and effective treatment with HCV cure rates of over 95% [14,15]. As PLWH have an increased risk of developing end-stage liver disease, all patients from this group should be vaccinated against HBV and HAV and treated for HBV and HCV [16,17,18]. The key to receive the necessary care and treatment to prevent or delay progression of liver disease and to decrease the further spread of HIV and viral hepatitis co-infection is the comprehensive screening of the population followed by preventive measures [19].

The aim of the study was to compare the testing and prevention of viral hepatitis among PLWH in the CEE region between 2019 and 2022.

## 2. Materials and Methods

The Euroguidelines in Central and Eastern Europe Countries (ECEE) Network Group was established in February 2016 to review the standards of care for HIV infection and viral hepatitis in the region. The Network consists of key experts from 24 countries of the region who are actively involved in HIV care [20]. Data was collected through two identical on-line surveys conducted in 2019 and 2022, built on the MonkeySurvey platform. The questionnaires consisted of questions collecting information about testing strategies, vaccination and/or treatment options of viral hepatitis A, B and C in CEE region. Detailed survey content is provided as a Appendix A.

### Ethical Aspects

The study was approved by the Bioethical Committee of the Medical University of Warsaw (reference number AKBE/61/2023), in full accordance with the Declaration of Helsinki and Good Clinical Practice. 

## 3. Results

A total of 18 of the 24 countries of the ECEE Network Group answered the questionnaire: Albania, Armenia, Belarus, Bosnia and Herzegovina, Bulgaria, Croatia, Czech Republic, Estonia, Georgia, Greece, Hungary, Lithuania, Poland, Romania, Slovakia, Serbia, Turkey, and Ukraine.

In 2019 Bosnia and Herzegovina, Romania, Serbia and Ukraine gave answers from two different clinics. In 2022 only Bosnia and Herzegovina gave answers from two different clinics; mostly the different clinics had the same/similar answers.

The overview of testing and treatment options in the CEE region are given in Table 1. Detailed information about testing for viral hepatitis A, B and C, and the treatment of hepatitis B and C are described in the following chapter.

### 3.1. Testing Strategies of HAV/HBV/HCV Coinfection

In 2019, in all 18 countries all HIV-positive patients were tested for HBV and HCV with the exception of one centre in Ukraine in 2019, where they recommended testing but, as the patients had to pay for the analysis, not all were tested. Testing of HBV and HCV was free of charge for the patients and covered mostly (72.2%) by health insurance or other governmental programs (22.2%); in Ukraine testing was available only as a paid service. 

In 2022 17/18 countries tested HIV-positive patients for HBV and HCV. Testing of HBV and HCV was free of charge for the patients and covered mostly (77.8%) by health insurance, other governmental programs (16.7%) or NGOs (5.6%). HBV and HCV testing principles are shown in Figure 1. 

In 2019, HBV and HCV were tested in 95.7% of new patients for the first time and in 72.7% of patients with an unexplained increase of hepatic transaminases; HBV was tested routinely once a year in 27.3% of patients, HCV in 45.5% of patients; in women who planned pregnancy or were pregnant HBV was tested in 45.5%, and HCV in 31.8% of patients; 77.2% of patients with risky behaviour got tested for HCV and 9.1% for HBV (mentioned under “others”); patients representing other principles were once every 6 months, once every 2 years, or once every 2–3 years.

In 2022, HBV and HCV were tested in 89.5% of new patients for the first time and in 79.0% of patients with an unexplained increase of hepatic transaminases; HBV was tested routinely once a year in 52.6%, and HCV in 42.1% of patients; in women who planned pregnancy or were pregnant HBV was tested in 57.9%, and HCV in 52.6% of patients; 63.2% of patients with risky behaviour got tested for HCV and 5.6% for HBV (mentioned under “others”); patients representing other principles were tested once every 6 months, once every 2 years, or once every 2–3 years.

In 2019, screening of HAV IgG was routine in 12/22 (54.5%) of participating clinics; however, in 2022, screening of HAV IgG was routine in 9/19 (47.4%) of participating clinics.

In 2019, 10/22 (45.5%) of participating clinics answered that they had tested all Hepatitis B surface antigen (HBsAg)-positive patients for hepatitis D. In 2022 7/19 (36.8%) of participating clinics confirmed that they had tested all HBsAg-positive patients for hepatitis D. 

### 3.2. Vaccination against Hepatitis A and B

In 2019, the vaccination of PLWH against hepatitis A was routine in 3/22 (13.6%) countries. The average price was 36 EUR. In the same year, in 6/22 (27.3%) clinics the vaccination against HAV was free of charge for HIV-positive patients.

In 2022, the vaccination of PLWH against hepatitis A was routine in 4/19 (21.5%) countries. The average price was 33 EUR, and in 5/19 (26.3%) clinics the vaccination against HAV was free of charge for PLWH.

In 2019, the vaccination of PLWH against hepatitis B was routine in 13/22 (59.1%) clinics, and free of charge in 11/22 (50.0%) clinics. The average price was 33 EUR.

In 2022, the vaccination of PLWH against hepatitis B was routine and free of charge in 9/19 (47.4%) clinics. The average price was 37 EUR. Detailed information about vaccination against hepatitis A and B is given in Table 2.

In terms of HBV treatment for HBsAg-positive or Hepatitis B surface antibody (HBsAb)-negative patients, the choice of NRTI was in 2019 and in 2022 tenofovir-based (either TDF or TAF) in 17/18 (94.4%) countries (one country had no answer). 

In terms of HCV treatment, all participating countries had access to DAA drugs. In 9/18 (50%) countries there were some limitations for DAA treatment access. The limitations by country are listed in Table 3. In 2019 there were many limitations, including enabling treatment only for those with advanced liver disease (fibrosis stage 3–4). There also was limited access for active IVDU, in the case of alcohol abuse and for migrants. The most common limitation in 2022 was that the patient had to have health insurance to get treatment, and also in some countries migrants did not get treatment. Only one country still allowed access to DAA treatment only for those with advanced liver disease.

Acute hepatitis C treatment was available in 2019 and in 2022 in 9/18 (50%) countries. Acute hepatitis C treatment was reimbursed in 2019 in 6/18 countries (33.3%), and in 2022 in 9/19 (50%) countries. HCV-infected patients could start treatment at the earliest 4 weeks–6 months after diagnosis. In 2019, 9/18 (50%) countries answered that HCV-infected patients had to wait 6 months to start treatment, in 2022 this was 5/18 (27.8%).

## 4. Discussion

This was a comparative study of testing, prophylaxis and treatment of viral hepatitis in PLWH in Central and Eastern Europe between 2019 and 2022. 

Comparing the screening of HBV and HCV coinfection in PLWH in the CEE region in 2019 and in 2022 to an earlier study where overall HBV and HCV screening polices in the CEE region were analysed, there was a considerable improvement as testing was more accessible than it was in 2017 [21]. Although the European AIDS Clinical Society (EACS) guidelines also recommend screening of HAV antibodies in all PLWH, in particular in MSM, as MSM are more likely to engage in high-risk sexual behaviour, the routine testing for HAV antibodies in the CEE region was still poor. The reason for inadequate HAV testing in the area could be associated partly with the dominant transmission routes but also with stigmatisation. In many countries, especially countries of the former Soviet Union, the predominant mode of HIV transmission has been historically intravenous drug use. Therefore, due to the shared routes of transmission, HBV and HCV are tested regularly in all PLWH, but as homosexuality in these countries is still more or less stigmatised, testing of HAV was insufficient [22].

Even though European EACS guidelines recommend that persons lacking anti-HAV IgG antibodies should be offered vaccination to prevent HAV infection regardless of their CD4+ T-cell count, vaccination against hepatitis A in the region is insufficient. Mostly, the vaccination of PLWH is routine in countries where the vaccination against HAV is free of charge for HIV-positive patients. It is much more difficult to evaluate vaccination against HBV as in many countries the HBV vaccine is a part of the national immunisation schedule, therefore most PLWH should already be vaccinated. Nevertheless, many countries vaccinate PLWH against HBV free of charge (50%); in other countries the vaccination fee varies from 10–150 EUR per dose, which could be an obstacle for some patients, mostly for those who need it the most, e.g., IVDU. Although vaccination against HAV and HBV is less immunogenic in HIV-infected patients than in the general population, immunization is strongly recommended by international and national guidelines, especially for PLWH [11,23]. However, vaccination coverage of HAV and HBV is low not only in the CEE region but also in Germany, France and the UK [23,24]. It is important to educate not only patients but also physicians about the vaccination to improve their knowledge and attitudes regarding the prevention of viral hepatitis [25]. Studies show that offering free vaccination and delivering targeted information by HIV specialists concerning vaccinations could be a good way to increase vaccine coverage in PLWH [26,27]. 

Previous studies showed that the uptake of DAA treatment in the CEE region has been modest, mostly due to very limited access to DAAs via healthcare systems [21,28]. Our current study shows remarkable improvement in HCV treatment. DAAs were available in all participating countries already in 2019 when the first survey was conducted, although regrettably, at the time only seven clinics had no limitations for treatment access. Although in most CEE countries participating in the survey there is a national hepatitis eradication programme, the survey conducted in 2022 shows that 50% of participating countries still have some limitations for DAA treatment access. Mostly the limitations are related to whether the patient has health insurance, as previously the restrictions were often related to the stage of fibrosis. In 2022, there were only two countries that still treat HCV only in the case of advanced liver disease. However, some limitations are still alarming: there are countries where active IVDUs or prisoners will not be treated, as well as people who abuse alcohol. In 2016, the World Health Organization (WHO) called for achieving the goal by 2030 with targets that included diagnosing 90% of all people with hepatitis C and treating 80% of those diagnosed with the virus. The WHO also recommended that treatment should be prioritised for PLWH who have HCV co-infection as they are at a higher risk of the rapid progression of liver fibrosis [29,30,31]. To reach the WHO elimination goals for 2030, it is important to treat everybody regardless of their accompanying barriers [29]. 

Latest EACS, European Association for the Study of the Liver (EASL), and American Association for the Study of Liver Diseases (AASLD) guidelines recommend that after initial diagnosis of acute HCV, treatment should be initiated without awaiting spontaneous resolution [32,33,34]. In the CEE region, only in 50% of countries is acute HCV treatment available and in even less (33%) is the treatment of acute HCV reimbursed by the government. Early treatment of acute HCV is especially recommended for PLWH as the spontaneous clearance of HCV is less likely to occur, and also for MSM with ongoing risk behaviour in order to prevent any further HCV transmissions [35,36]. There have been a number of published studies assessing the strategy of shorter durations of HCV therapy in patients with acute and early HCV infection, and there are currently ongoing studies to assess the efficacy of a shortened duration of acute HCV treatment, but to avoid the potential for relapse, for now standard regimens, the same as for chronic HCV infection, are recommended [37,38].

Some limitations should be noted regarding our study. First, as mentioned above this study was conducted as an online survey and not all the countries from the ECEE Network Group participated in the survey. Secondly, different clinics from the same country had different approaches to the treatment and prophylaxis of viral hepatitis. Thirdly, as the ECEE Network Group consists only of countries from the CEE region, we do not have data on how the levels of screening/treatment compare to PLWH in Western Europe or the United States. Despite these limitations, we believe that this study gives an adequate overview of testing, prophylaxis and treatment strategies in the CEE region.

## 5. Conclusions

We have made the following observations: screening of HBV and HCV coinfection in PLWH in the CEE region was very good, both in 2019 and in 2022.; screening of HAV antibodies in the CEE region was poor in 2019 and in 2022; vaccination against HAV in the region is insufficient, and at the same time vaccination of PLWH against HBV was routine and free of charge in 50% of participating clinics both in 2019 and 2022; all participating countries have DAA drugs available and the limitations in access to treatment have decreased over the compared years; and acute HCV treatment access did not change, although the reimbursement options decreased from significantly to moderately narrowed. 

Our study shows that vaccination against HBV and especially against HAV has room for improvement. Although the use of DAAs against HCV among PLWH in CEE in general has improved, there are still barriers that need to be overcome to achieve the WHO goals for 2030.

## Figures and Tables

**Figure 1 vaccines-11-00980-f001:**
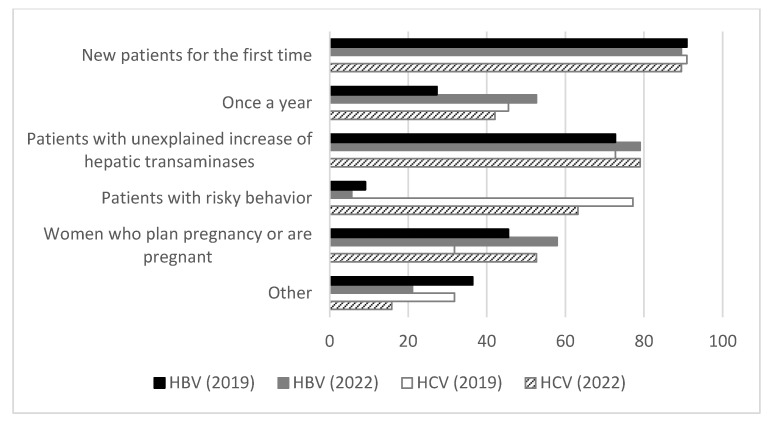
The frequency and principles of HBV and HCV screening in HIV-positive patients 2019 and 2022. Black bars indicate HBV testing in 2019, grey bars indicate HBV testing in 2022, white bars indicate HCV testing in 2019, striped bars indicate HCV testing in 2022.

**Table 1 vaccines-11-00980-t001:** Viral hepatitis testing and treatment options in CEE in 2019 and 2022.

	2019 (*n* = 22)	2022 (*n* = 19)
	yes	yes
Does your clinic screen all HIV-positive patients for HCV Ab?	21 (95.5%)	18 (94.7%)
Is HCV testing free of charge in your country?	20 (90.9%)	19 (100%)
Does your clinic screen all HIV-positive patients for HBsAg?	21 (95.5%)	18 (94.7%)
Is HBV testing free of charge in your country?	20 (90.9%)	19 (100%)
Does your clinic screen all HIV-positive patients for HAV IgG?	12 (54.5%)	9 (47.4%)
Does your clinic screen all HBsAg-positive persons for hepatitis Delta?	10 (45.5%)	7 (36.8%)
Do you have an access to DAA drugs in your country?	22 (100%)	19 (100%)
Is acute hepatitis C treatment available in your country?	12 (54.5%)	10 (52.6%)
If yes, is acute hepatitis C treatment reimbursed in your country?	7 (31.8%)	10 (52.6%)

Note: *n*—the number of participated clinics; HCV Ab—Hepatitis C virus antibody; HBsAg—Hepatitis B surface antigen; HAV IgG—Hepatitis A virus IgG antibody; DAA—direct-acting antiviral.

**Table 2 vaccines-11-00980-t002:** Strategy of vaccination against hepatitis A and B in years of 2019 and 2022.

	2019 (*n* = 22)	2022 (*n* = 19)
Do you vaccinate all HIV-positive patients against HAV? (Yes)	3 (13.6%)	4 (21.55%)
Are vaccinations against HAV free of charge in your country? (Yes)	6 (27.3%)	5 (26.3%)
If no, what is the average price (in EURO) of HAV vaccination?	15–120 EUR (36.1 EUR)	10–80 EUR (33.3 EUR)
Do you vaccinate all HIV-positive patients against HBV? (Yes)	13 (59.1%)	9 (47.4%)
Are vaccinations against HBV free of charge in your country? (Yes)	11 (50.0%)	9 (47.4%)
If no, what is the average price (in EURO) of HBV vaccination?	10–150 EUR (33.1 EUR)	10–130 EUR (36.8 EUR)
Do you have your own national recommendations regarding vaccinations against HAV and HBV? (Yes)	18 (81.8%)	16 (84.2%)
Does your clinic have a protocol for reviewing vaccine response and re-vaccinating where needed? (Yes)	12 (54.5%)	7 (36.8%)

Note: HAV—Hepatitis A virus; HBV—Hepatitis B virus. *n*—the number of participating clinics.

**Table 3 vaccines-11-00980-t003:** Limitations for DAA public treatment access.

Country	Limitations	
	2019	2022
Albania	only for F3–F4	only F3–F4
Armenia	not available free of charge, most patients cannot afford it	no limitations mentioned
Bosnia and Herzegovina	limited number of patients get treatment per year	health insurance
Bulgaria	no limitations mentioned	health insurance
Croatia	active IVDU	no limitations mentioned
Estonia	only F2–F4	health insurance
Georgia	no limitations mentioned	migrants (citizens of Ukraine are an exception)
Hungary	health insurance	no limitations mentioned
Romania	only F1–F4, health insurance, alcohol abuse, active IVDU	no limitations mentioned
Serbia	only F3–F4	only F3–F4
Slovakia	active IVDU, alcohol abuse	active IVDU, health insurance, migrants
Turkey	migrants, health insurance	unregistered migrants, health insurance
Ukraine	very limited access	no limitations mentioned

Note: F—fibrosis stage; IVDU—intravenous drug use.

## Data Availability

Data supporting the reported results can be obtained from the corresponding author upon request.

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
