# Peer review of "Vaccination against HBV and HAV as Mode of Hepatitis Prevention among People Living with HIV—Data from ECEE Network Group"

_vaccines, 2023, doi:10.3390/vaccines11050980_

Round 1

Reviewer 1 Report

Review of Vaccination against HBV and HAV as mode of hepatitis prevention among people living with HIV – data from ECEE Network Group by Aimla et al.

This is a straight forward paper looking into the rates of vaccination for co-infectious viral diseases in patients living with HIV. Surprisingly, it appears that Hepatitis A (HAV) screening is not as common as other hepatitis (C and B). Overall the study design and approach makes sense and the reviewer found the paper informative. It would be interesting to see (from the reviewer’s perhaps bias standpoint) how the levels of screening/treatment compare to HIV patients living in western Europe or United States. Could this data be provided for a clinic in either for such to perhaps serve as benchmark of HAV vaccinations to aim for? It is hard to kind of warp your mind around if say 16.7% vaccination rate is good, average, bad or horrible… 

Author Response

Dear Reviewer, thank you, we appreciate your very thoughtful comment!

This is a straight forward paper looking into the rates of vaccination for co-infectious viral diseases in patients living with HIV. Surprisingly, it appears that Hepatitis A (HAV) screening is not as common as other hepatitis (C and B). Overall the study design and approach makes sense and the reviewer found the paper informative. It would be interesting to see (from the reviewer’s perhaps bias standpoint) how the levels of screening/treatment compare to HIV patients living in western Europe or United States. Could this data be provided for a clinic in either for such to perhaps serve as benchmark of HAV vaccinations to aim for? It is hard to kind of warp your mind around if say 16.7% vaccination rate is good, average, bad or horrible…

As the Euroguidelines in Central and Eastern Europe Countries Network Group consists only of countries from CEE region we don’t have data how the levels of screening/treatment compare to PLWH in Western Europe or United States. We added this issue to limitations. (Corrected manuscript attached)

Reviewer 2 Report

The study is of interest and the study design is appropriate. The topic is relevant and the conclusion are important. I only few comments:

- please make sure to spell abbreviations the first time you use them, even in the abstract and keep it consistent (HAV, HBV spelled HCV is not, not DAAs)

- please make sure to place references at the proper place (for example line 59 and 62 in the introduction)

- It would have been really nice to have a comparison with available data in the rest of Europe. I do understand that this study is focusing on the selected countries but it completely misses to place in the context of the rest of Europe/rest of the world. If these data don't want to be mentioned at least should be stated in the limitation of the study

Author Response

Dear Reviewer, thank you for drawing our attention!

- please make sure to spell abbreviations the first time you use them, even in the abstract and keep it consistent (HAV, HBV spelled HCV is not, not DAAs)

- please make sure to place references at the proper place (for example line 59 and 62 in the introduction)

We have made the corrections in manuscript. It’s traceable with „Track Changes“ (corrected manuscript attached)

- It would have been really nice to have a comparison with available data in the rest of Europe. I do understand that this study is focusing on the selected countries but it completely misses to place in the context of the rest of Europe/rest of the world. If these data don't want to be mentioned at least should be stated in the limitation of the study

Thank you! We added this issue to limitations: As the Euroguidelines in Central and Eastern Europe Countries Network Group consists only of countries from CEE region we don’t have data how the levels of screening/treatment compare to PLWH in Western Europe or United States.